# Comparison between Macro and Trace Element Concentrations in Human Semen and Blood Serum in Highly Polluted Areas in Italy

**DOI:** 10.3390/ijerph191811635

**Published:** 2022-09-15

**Authors:** Aldo Di Nunzio, Antonella Giarra, Maria Toscanesi, Angela Amoresano, Marina Piscopo, Elisabetta Ceretti, Claudia Zani, Stefano Lorenzetti, Marco Trifuoggi, Luigi Montano

**Affiliations:** 1Department of Chemical Sciences, University of Naples Federico II, Via Cinthia, 21, 80126 Naples, Italy; 2Istituto Nazionale Biostrutture e Biosistemi-Consorzio Interuniversitario, Viale delle Medaglie d’Oro, 305, 00136 Rome, Italy; 3Department of Biology, University of Naples Federico II, 80126 Napoli, Italy; 4Department of Medical and Surgical Specialties, Radiological Sciences and Public Health, University of Brescia, 25123 Brescia, Italy; 5Department of Food Safety, Nutrition and Veterinary Public Health, Italian National Institute of Health (ISS), 00161 Rome, Italy; 6Andrology Unit and Service of Lifestyle Medicine in UroAndrology, Local Health Authority (ASL) Salerno, Coordination Unit of the Network for Environmental and Reproductive Health (EcoFoodFertility Project), Italy “Oliveto Citra Hospital”, Oliveto Citra, 84020 Salerno, Italy; 7PhD Program in Evolutionary Biology and Ecology, University of Rome Tor Vergata, 00133 Rome, Italy

**Keywords:** trace elements, ICP-MS, semen quality, male fertility, dietary intake, environmental exposure

## Abstract

Macro and trace elements are important regulators of biological processes, including those ones linked to reproduction. Among them, Ca, Cu, Fe, K, Mg, Mn, Na, Se, and Zn ensure normal spermatic functions. Hence, the aim of this study was to evaluate the concentrations of 26 macro and trace elements (Al, As, Ba, Be, Ca, Cd, Co, Cr, Cu, Fe, Hg, K, Li, Mg, Mn, Na, Ni, Pb, Rb, Sb, Se, Sn, Sr, U, V, and Zn) in blood serum and also in semen of healthy young men, homogeneous for age, anthropometric characteristics, and lifestyle, living in three highly polluted areas in Italy. Furthermore, a comparison among three geographical areas was performed to highlight any difference in the investigated parameters and, overall, to speculate any correlations between chemical elements and semen quality. Statistically significant differences (*p* < 0.05) among the three areas were found for each investigated element, in both semen and serum samples, where inter-area differences were more evident in semen than in blood serum, suggesting human semen as an early environmental marker. Considering the homogeneity of three cohorts, these differences could be due more to environmental conditions in the recruiting areas, suggesting that variations in those involved in reproductive-associated pathways can have an impact on male fertility. Nevertheless, more research is needed to evaluate threshold values for sperm dysfunction and male infertility. Actually, the role of different dietary intake and environmental exposure underlying the observed differences in the recruiting areas is under further investigation for the same cohort.

## 1. Introduction

Varying concentrations of different minerals, known as trace elements, are found in the body and are essential for maintaining normal growth and development of living tissues [1,2,3]. Examples of essential trace elements include minerals such as Iron (Fe), chromium (Cr), copper (Cu), zinc (Zn), selenium (Se), molybdenum (Mb), cobalt (Co), and iodine (I). While these elements only account for 0.01–0.02% of the total body weight, they play significant roles such as enzyme catalysts or in oxidation–reduction reactions for energy metabolism [4]. Iron in particular is a vital component in haemoglobin and myoglobin for the transport of oxygen [5].

Thus, the necessity or requirement to set a level for daily intake arises to meet physiological needs and, thereby, to reduce the risk of nutrient deficiency or excess [4,6].

All trace elements can be potentially toxic if consumed at high levels for prolonged periods, which can result in impairment of normal physiological functions and/or the development of pathological conditions [7,8].

The potential for trace elements to act as toxicants on human health is of particular interest due to their extensive use in various industrial processes, which can result in their release into the environment, causing bioaccumulation and poisoning within our bodies. Trace elements found with increasing frequency in nature with recognised harmful impacts on living beings include elements such as arsenic (As), lead (Pb), and mercury (Hg) [9].

The toxicity of trace elements is correlated with several factors, such as age at exposure, gender, and capacity for biotransformation, all of which are host-based factors that can affect the toxicity mechanism [10]. In addition, lifestyle factors, such as smoking or alcohol ingestion, can affect the level of metal intoxication or exposure. Cigarette smoke itself contains many toxic metals [11], such as cadmium, and is considered the main source of cadmium exposure [12]. 

In recent years, the incidence of human male reproductive disorders has been increasing worldwide [13]. An extensive number of studies have suggested that this decline in semen quality may be due to exposure to environmental pollution [14,15]., The increased number of fertility disorders is potentially correlated with the widespread exposure of the general population to metalloid and metal species [16,17]. For instance, excessive exposure to an essential element such as Cu(II) may lead to oxidative damage, with adverse effects on spermatogenesis [18,19]. Recently, the existence of copper-dependent oxidative DNA damage was demonstrated, triggered by the recognition by Cu(II) of the arginine residues of sperm H1 histones, and not somatic H1 histones, providing a new insight into the mechanisms of copper toxicity [20].

Oxidative stress conditions can affect human sperm function and quality, even though reactive oxygen species (ROS) are needed to regulate normal sperm physiology [21,22]. Hence, human semen is seen as an early and sensitive biomarker of environmental exposure to pollutants, which may lead to a better understanding of how environmental toxicants can affect healthy humans [16,23,24].

The aim of this study was to evaluate blood serum and semen levels for trace elements in healthy young men homogeneous for age, anthropometric characteristics, and lifestyle, living in three highly polluted areas in Italy, and to determine how the environmental conditions of the recruitment areas influence the bioaccumulation of these trace elements. The concentrations of 26 chemical elements (Al, As, Ba, Be, Ca, Cd, Co, Cr, Cu, Fe, Hg, K, Li, Mg, Mn, Na, Ni, Pb, Rb, Sb, Se, Sn, Sr, U, V, and Zn) were determined by inductively coupled plasma mass spectrometry (ICP-MS) in human blood serum and semen to evaluate any differences in bioaccumulation in the two fluids, and to consider any correlations between chemical elements and semen quality.

Subjects recruited for this study were the participants of the randomized controlled trial titled “Effects of Lifestyle Changes on Semen Quality in Healthy Young Men Living in Highly Polluted Areas”, a FAST study (registered on ClinicalTrials.gov Protocol Registration and Results System (PRS) n. J59D1600132001) [25] whose detailed data in results obtained on the semen quality are given in Montano et al. (2021) [26].

## 2. Materials and Methods

### 2.1. Reagents and Materials

All reagents used were of analytical grade and for ultra-trace analysis (Sigma Aldrich, Ultrascientific, Merck). Certified reference materials (CRMs) were provided by the European Commission, Joint Research Centre, Institute for Reference Materials and Measurements (IRMM).

### 2.2. Ethical Statements

All methods were performed in accordance with the guidelines and regulations contained in the Declaration of Helsinki [27].

All experimental protocols were approved by the Ethics Committee of Southern Campania with protocol number 325 (29 November 2017) and by that of Brescia Province with protocol number 2980 (13 March 2018),and were accepted by the Italian National Institute of Health (20 December 2017). 

The FAST study was registered on ClinicalTrials.gov with NTC number NCT04012385 [25]. All recruited donors signed their informed consent before their visit with an andrologist and sample collection.

### 2.3. General Description

A detailed description of the population studies, recruitment and inclusion/exclusion criteria, and sample collection and processing are described in Montano et al. (2021) [26]. In brief, the recruitment was conducted from April 2018 to January 2019 as part of the enrolment of the FAST study “Effects of Lifestyle Changes on Semen Quality in Healthy Young Men Living in Highly Polluted Areas” [25].

All recruited subjects were informed of the study objectives and gave their signed consent before sample collection. At enrolment, upon andrological and nutritional visits (data were collected by questionnaire), a blood sample and a semen sample were taken from 323 young healthy men aged between 18 to 22 years old. In addition, other inclusion criteria were normal weight and waist circumference, normal sperm count, daily alcohol intake of less than 5 alcohol units a week, daily smoking of less than 5 cigarettes, and having lived in one of the recruitment areas for at least 5 years.

The Italian geographical areas chosen for subject recruitment are located between the National Priority Sites (SIN, so called “National Interest Sites”), which represent very large, polluted areas classified by the Italian State and which require remediation of the soil, subsoil, and underground and surface waters to avoid environmental and health damage and areas involved in illegal waste dumping practices [28]. The areas selected for the recruitment are shown in Appendix A. 

Samples from the first site came from the SIN called Brescia-Caffaro (BSC, n = 142), located in the city of Brescia, a highly industrialized city in the Lombardy Region (northern Italy); this area has a high level of environmental pollution, caused by the activities of a chemical factory, which produced Polychlorinated Biphenyls (PCBs) and PCB mixtures, such as Fenclor and Apirolio, between 1930 and 1984 [29,30].

Samples of the second area came from the Sacco River Valley (SRV, n = 54), Lazio Region (central Italy), located near Rome, which is heavily polluted by industrial waste from a chemical industrial plant, contaminated by beta-hexachlorocyclohexane (β-HCH), a persistent organic compound belonging to the group of hexachlorocyclohexane isomers [31].

Samples from a third area came from the municipalities belonging to the “Land of Fires” (LF, *n* = 127), a highly polluted area between the northern city of Naples and Caserta, Campania Region (Southern Italy), known for illegal waste dumping and toxic fires [32]. Many studies have focused on the environmental illegal waste problem in this area, which causes negative effects on human health [33].

### 2.4. Semen and Serum Collection

Semen and blood serum samples were collected at the structures of the Brescia Hospital, “Medicina Futura” Center of Acerra, “Villa dei Fiori” Clinic of Acerra and AVIS sections of Ferentino and Frosinone.

Semen samples were collected in a sterile container through masturbation, after a period of abstinence from sexual activity of 3 to 5 days. A portion of 600 µL was frozen within 30–40 min after collection and delivered to the laboratory for the quantification of trace elements. 

Peripheral venous blood samples were collected using metal-free needles and syringes, and, after clot formation, serum was obtained through centrifugation at 3000 rpm for 15 min at room temperature. A 1 mL aliquot of serum was transferred into a metal-free vial, frozen, and delivered to the laboratory for the determination of trace elements.

### 2.5. Macro and Trace Elements Analysis

To evaluate differences in blood serum and semen, we grouped detected chemical elements into five groups according to the classification used by WHO, which considers element nutritional importance and its daily requirements [1,3]. The elements of Group I were not the object of this study. The elements of Groups II, III, IV, and V are briefly described below. 

Group II was made up of sodium (Na), potassium (K), magnesium (Mg), and calcium (Ca), which are also called “macro elements”. These elements play an important nutritional role, and their daily intake is more than 100 mg/day.

Group III was made up of essential trace elements, such as copper (Cu), iron (Fe), manganese (Mn), cobalt (Co), nickel (Ni), selenium (Se), and zinc (Zn). Their daily intake is less than 100 mg/day, but a deficiency could cause disorders and might be fatal [1,34].

Group IV was made up of additional elements that includes elements with a not well-defined role but which are most likely essential, such as arsenic (As), cadmium (Cd), chromium (Cr), lithium (Li), rubidium (Rb), strontium (Sr), and vanadium (V). A requirement of essentiality for an element is related to its role in biochemical functions. Hence, an element could be defined as essential when the lowering of its intake results in a strong reduction of a physiologically important function or when it is a constituent part of an organic structure involved in a vital function [3].

Group V was made up of other non-essential elements that have an unknown role and which could be toxic, such as aluminium, antimony (Sb), barium (Ba), beryllium (Be), mercury (Hg), lead (Pb), and Tin (Sn) [1,34].

The listed elements were evaluated both in blood serum and semen samples. Iron was evaluated only in semen samples, because it was shown that high dietary Fe intakes can lower sperm concentration and motility [35]. For determination of Al, As, Ba, Be, Ca, Cd, Cr, Co, Cu, Fe, K, Li, Mg, Mn, Hg, Na, Ni, Pb, Rb, Sb, Se, Sn, Sr, V, and Zn, 500 µL of semen or blood serum was digested with 1 mL of HNO_3_ ≥ 69% (*v/v*) in a glass vessel in a microwave system equipped with an autosampler for 5 min at 160 °C (DISCOVER SP-D, CEM, Bergamo, Italy); after cooling, the obtained solution was taken to a final volume of 10 mL with a solution of HNO_3_ ≥ 2% (*v/v*). 

After acid digestion, the samples were analysed by microwave plasma optical emission spectrometry (MP-AES 4210, Agilent, Santa Clara, CA, USA) for the determination of Ca, K, Mg, and Na.

The samples were tested by inductively coupled plasma mass spectrometry (ICP-MS, Aurora M90 Bruker, Bremen, Germany [19] for the determination of the other mentioned elements.

The difference in the number of analysed samples was because it was not possible to perform all determinations on all samples. 

A blank from collecting tubes was evaluated for the possible presence of detected elements. A blank digestion of 1 mL of HNO_3_ ≥ 69% (*v*/*v*) in a glass vessel was performed to detect metal contamination for each digestion batch of 20 samples.

After the stabilization of the instruments, a calibration curve for each element was performed every analysis session, calculated on five concentration standard solutions obtained from certified standard solutions. After the calibration procedure, a standard sample was run at the start and every 20 samples in the analytical lot to verify instrument calibration. The results of the test samples were reported considering the values obtained for each element in the blank samples. The limit of detection (LOD) and limit of quantification (LOQ) were calculated by method of blanks variability for each investigated metal. The LOD and LOQ values for each element are shown in Appendix A. The limit of detection (LOD) and limit of quantification (LOQ) in the final sample are expressed in µg/L.

Trace element measurement precision was estimated by performing at least seven replicates on an unfortified blood serum sample. The sample was then fortified by known additions of elements not contained in the unfortified sample. The obtained data are shown in Appendix A. Precision, accuracy, and recovery data were evaluated for the blood serum samples.

The accuracy of the method was evaluated by analysing two different CRMs of human serum. Mg, Ca, and Li were evaluated with CRM BCR-304 (IRMM). Zn and Se were analysed in CRM BCR-638 (IRMM). Detailed data are shown in Appendix A. Recovery data were evaluated for the CRMs. 

The accuracy for the elements not contained in the CRMs was evaluated by replicates of the fortified sample. These data are reported in Appendix A. Precision, accuracy, and recovery data were evaluated for the blood serum samples.

Data of precision and accuracy of the described method were in accordance with the requirements reported in ISS Report 15/30 [36], in which a recovery between 80 and 120% was considered acceptable and values of CV% in the range from 16 to 35% were reported for the blood serum matrix.

Precision and accuracy were estimated for semen samples by performing three replicates on an unfortified semen sample. The sample was fortified by known additions for evaluating precision data for the elements not contained in unfortified sample. In addition, recovery was evaluated by three replicates of the fortified sample. The obtained data are available in Appendix A. Precision, accuracy, and recovery data were evaluated for the semen samples.

### 2.6. Data Analysis

All the data of semen and serum analyses were collected in a database with Microsoft Excel. Data analysis was carried out using the Stata 14.2 software (Stata Corp, College Station, TX, USA). 

First, an analysis of the distribution of all detected trace elements in the serum and semen of the donors of the whole cohort and the three subgroups living in the three different areas was performed to evaluate median values and the range of the values’ distribution. 

The Kruskal–Wallis test was carried out—for each element of the three subgroups and in both semen and blood serum—to evaluate whether the samples originate from the same distribution. The Kruskal–Wallis test was performed when an element was below the LOD in less than 70% of the samples for each specific matrix. In this case, the analysis was carried out on the values greater than the LOD. No analysis was performed when an element was below the LOD in more than 70% of the samples.

The percentage of serum and semen samples below the LOD is shown in Appendix A. Blood serum samples below the limit of detection (LOD) are expressed in % in Appendix A. Semen samples below the limit of detection (LOD) are expressed in % in Appendix A.

## 3. Results

The general characteristics of the young men recruited from the three polluted areas were similar in terms of body weight, height, BMI, and abdominal circumference (Appendix A); the BMI in some boys was higher than the limit of 25 for normal weight, but in this case, they were subjects with particularly developed muscles, and it was not due to abdominal obesity as assessed with the measurement of abdominal circumference. Only the average age was higher in the BSC group (Appendix A).

Semen parameters including semen volume, pH value, spermatic concentration, total motility, and sperm morphology of the subject groups were previously reported and discussed in Montano et al. (2021) [26]. A short overview is provided in Appendix A. Lifestyle factors (PREDIMED and IPAQ scores) and semen quality parameters of subjects in each area are reported as mean value ± standard deviation.

Table 1 displays the median and range values for blood serum and semen of the macro elements Ca, Mg, Na, and K in the whole cohort and in the subgroups of the three areas investigated in this study. There was a statistically significant difference (*p* < 0.05) in semen for every element among the areas. In semen samples, the BSC group had the highest values of Ca, Mg, Na, and K compared with LF and SRV, while SRV had the lowest values. In blood serum samples, the BSC subgroup had the lowest values compared with LF and SRV, while SRV had the highest values.

Table 2 displays the median and range values for blood serum and semen of the essential trace elements Cu, Fe, Mn, Ni, Se, and Zn in the whole cohort and in the subgroups of the three areas investigated in this study, except for iron, which was not considered for the serum matrix. There were statistically significant differences (*p* < 0.05) among the three areas for every trace element. In semen samples, the BSC subgroup had the highest values compared with LF and SRV, and also for these trace elements. The values of Fe, Ni, and Zn were from two- to five-fold higher than in SRV and LF. The content of Mn, Se, and Zn was comparable for LF and SRV. In blood serum samples, the LF subgroup had the highest values of Cu, Mn, and Se, while Zn was comparable for BSC and LF subgroups. The SRV subgroup had the lowest values of these five essential trace elements. Ni concentrations are not shown in the Table 2, as they were under the LOD value in all serum samples for the whole cohort and for each of the three subgroups.

Additional trace elements and other non-essential ones that were taken into account within this study were Al, As, Ba, Be, Cd, Co, Cr, Hg, Li, Pb, Rb, Sb, Sn, Sr, U, and V. Table 3 displays the median and range values for blood serum and semen of these elements with values above the LOD value for the whole cohort and for each of the three subgroups. In semen samples, Al, Be, Cd, Co, Cr, U, and V showed concentrations below the LOD for the whole cohort and for each of the three subgroups, while in blood serum samples, only Al, Be, Co, Cr, and V were below the LOD value in all the areas.

Statistically significant differences (*p* < 0.05) among the three areas were found for all the elements for which the Kruskal–Wallis test was conducted.

In semen samples, Hg and Sb were detectable and had comparable values in BSC and LF groups, while they were below the LOD value in the SRV group. LF presented values of As, Li, and Pb from two- to four-fold higher than BSC and SRV. LF and SRV subgroups were characterized by higher Ba concentration, approx. two-fold higher than BSC.

In blood serum samples, Cd was below the LOD for the BSC subgroup, while the LF subgroup showed a concentration of 0.7 µg/L, approx. two-fold higher than SRV. Sb, detected only in the BSC and LF subgroups in semen samples, was present above the LOD in serum for all three groups; however, the BSC subgroup presented the highest concentration. 

As shown in semen, the BSC and LF subgroups had comparable values of Hg, which were below the LOD in the SRV subgroup. We can see an opposite trend for Ba in blood serum, with the highest value for the BSC subgroup, approximatively two-fold higher than LF and SRV. Li and Pb showed higher concentrations in the BSC subgroup than in LF and SRV. 

In order to obtain an easier graphical representation of data comparison of the measured elements in the three different subgroups, it was decided to also report the concentrations of each element as percentage relative concentrations using the following formula:(1)$${\left[\%C\right]}_{M}=\frac{{\left[C\right]}_{M}\times 100}{{\left[C\right]}_{H}}\;$$
where the [%*C*]*_M_* values are calculated within each element for each subgroup (BSC, LF, SRV). The subscript *M* identifies the subgroup under examination for that element, while the subscript *H* represents the subgroup with the highest concentration for that element.

The comparison of data in blood serum—for each element calculated with (1)—is reported in Figure 1.

The comparison of data in semen among the population subgroups (BSC, LF, SRV) of this study, calculated with (1) to be expressed as percentage relative concentrations, is reported for each element in Figure 2.

Figure 2a clearly highlights the BSC subgroup as the one with the highest concentration of all macro and essential trace elements in semen.

## 4. Discussion

This study provides a baseline composition of some trace elements in human blood serum and semen in a young healthy Italian male population. The recruited cohort [26] was formed by 18–22-year-old individuals of three different subgroups homogeneous for anthropometric characteristics and lifestyle living in three distinct highly polluted areas of three Italian regions (Lombardy, Lazio, and Campania). The comparative analysis conducted among the population subgroups of this study (BSC, LF, SRV) highlights differences that may be correlated to different dietary intake and environmental exposures. The number of participants, although sufficient to highlight statistically significant differences, could represent a study limitation; in particular, it is rather small in the SRV area. This was due to the difficulty of finding students, all volunteers, that met the inclusion criteria of this study. Moreover, in this study, we have not considered unpolluted areas because we already reported that in the semen from the “high impact” (HI) group (in Campania areas), higher zinc, copper, chromium, and reduced iron levels; reduced sperm motility; and higher sperm DNA Fragmentation Index (DFI) in comparison with the “low impact” group (LI) was observed. Overall, several semen parameters (reduced sperm quality and antioxidant defences, altered chemical element pattern) were shown to be associated with residence in a highly polluted environment [16] and, in particular, in the SRV area, in which we recently found molecular alterations and severe abnormalities in the spermatozoa of young men [37].

Blood serum data of the present study were compared to proposed Reference Values of SIVR (Italian Society of Reference Values) [38] and Italian biomonitoring data of the Italian National Institute of Health (ISS) [39,40], and also to the data of WHO [3,41] listed in Appendix A. Reference Values and biomonitoring data are expressed as µg/L for macro and essential trace elements in blood serum in Appendix A. Reference Values and biomonitoring data are expressed as µg/L for additional and non-essential trace elements in blood serum. 

Our findings show values comparable to those mentioned above for most of the macro and essential trace elements listed in Appendix A, except for Mn, which showed concentrations higher than the reference values in the LF and BSC subgroups.

In addition, most of the additional and non-essential trace elements were found at concentrations in agreement with corresponding reference values reported in Appendix A, except for Ba, Li, Pb, and Sb, which showed concentrations higher than the reference values. Cd was present above the LOD in the LF and SRV subgroups, with a concentration comparable to the highest reference value.

With regard to the BSC subgroup, industrial pressure in the city of Brescia and its province has caused heavy metal pollution, particularly by Mn, Pb, and Ni [42,43,44,45]. The “Land of Fires” (LF subgroup) has been extensively studied for its pollution—due to massive landfills and illegal fires—and the high incidence of cancer [46,47,48,49].

However, trace element content is profoundly influenced by the consumption of food and drinking water, particularly cereal and cereal-based foods and bottled drinking water [50]. In addition, Naples and its province present a typically volcanic profile; therefore, it is not uncommon to find geological formations containing elements such as As and Mn, which could be a source of metals for water supplies used for drinking and irrigation purposes [51,52]. 

The SRV area is notorious for its high level of industrial mistreatment of the area and the resulting pollution mainly due to β-hexachlorocyclohexane, as well as being involved in illegal waste trafficking [31,53,54].

Nowadays, acute metal poisoning is still common in many developing countries. In addition, food and beverages, including drinking water and wine [55,56], can be sources of lead exposure, while vegetables farmed near really congested roads contain increased levels of Pb [55].

Some elements, such as Ba and Mn, were reasonably also traceable to industrial activities [43]. Although barium is not considered a nutritive essential element, people are exposed to this metal primarily by ingestion of food and water and inhalation of ambient air. The Italian Ba level in tap water ranges from 0.1 to 5000 µg/L, showing high variability due to the different kinds of soils [52]. In addition, the dietary intake of Ba can be increased because some plants—grown in Ba-rich soils—can accumulate high levels of this element [57,58,59,60]. Moreover, in 1977, it was estimated that—only with soluble Ba salts—up to 75% of inhaled Ba could be absorbed into the bloodstream [61]. This could explain why, in our study, Ba levels were higher in blood serum samples compared to reference values for the whole cohort, with a higher concentration in the BSC group than in LF and SRV. Our findings show a different trend in semen samples, with a concentration up to three times higher in LF and SRV subjects compared to those of BSC. This could suggest that semen might be one of the preferential accumulation districts of barium, showing a high exposure to this metal which was not noticed by the differences in the blood serum.

Arsenic—despite its known toxic effects in humans—is largely used in both industry and agriculture. Toxic exposure to arsenic has increased with the intensified consumption of the metalloid. Several As-based pesticides have been used in agriculture [62]. Despite the ban of inorganic arsenic, the past use of pesticides brought about elevated levels of arsenic and lead in soils [63], while organic arsenic compounds are still used as herbicides today [64]. Harmless organic forms of arsenic may be contained in fish and seafood [55,56]. The reduction of antioxidant pools and the arsenic-induced formation of free radicals show—as a net effect—a increased oxidative stress of cells [65,66,67,68].

It is worth noting that—in semen—LF subjects showed the highest value for lithium. This element can be found in varying amounts in foods. Italian tap water values of Li range from 5.1 to 60.8 µg/L [51]. Considering this data, a vegetarian diet—rich in grains and vegetables—would provide more Li compared to an animal protein-rich diet. In fact, Li content—such as that of other elements—in plants depends on its content in the surrounding environment [69,70]. Furthermore, it is well documented in the scientific literature how some elements are affecting and/or contributing to some biological parameters [71]. Zinc plays a critical role in spermatogenesis and stabilizing sperm cell membrane and nuclear chromatin [72]. The numerous roles found for different elements are summarized in Appendix A. The elements from Ca to Zn can be defined as essential to semen because they are deeply involved in several spermatic functions, and their possible deficiencies can bring about pathological status. In contrast, elements As to V are not known to have a defined role in sperm function, but some studies on how they influence sperm parameters have been reported [73,74,75,76,77,78,79,80,81,82].

As shown in Figure 2, the BSC subgroup shows the highest values for the elements strictly involved in all the semen functions. High concentrations of Zn, Fe, and Se are correlated with high spermatic concentrations, while decreased concentrations of these elements—as we noted in LF—are correlated with a lowered spermatic count [71,76,83,84,85,86].

These discoveries are in accordance with the baseline evaluation of the quality parameters of the subject reported in Montano et al. 2021 [26], in which the BSC subgroup showed the highest sperm concentration, while LF donors showed the lowest.

Nevertheless, it was shown that high dietary Fe intakes can lower sperm concentration and motility [35]. Moreover, high values of barium could be related to sperm morphology and motility [74]. SRV donors displayed the highest values of Ba coupled with the lowest values of copper. In addition, this evidence is in agreement with spermiogram analysis, which showed the lowest percentage of cells with normal morphology and the lowest total and progressive motility in SRV donors compared to the other groups [26].

However, our data show that some pollutants were present in both matrices, and some of them—such as Li and Mn—were in higher concentrations in semen, which led us to speculate that semen may be an internal site of bioaccumulation. 

Cu and Se, which showed higher concentrations in blood serum than semen in the three subgroups, might reflect a metabolic imbalance traceable to living conditions. In addition, zinc and copper are strongly related in humans, and their interactions are primarily antagonistic. Under physiological conditions, it has been shown that a constant proportion exists between these two trace elements in blood serum; the ratio of Zn/Cu ranges from 0.9 to 1.27. This ratio would provide useful information on semen quality. Indeed, it has been reported that, among infertile men, blood serum zinc decreased while the copper value increased, and the value of Zn/Cu was lower than that of fertile men [87].

In our research, we examined the blood serum Zn/Cu ratio for the whole recruited cohort and the three subgroups, as displayed in Appendix A. In the three subgroups we noticed an increase in blood serum Zn/Cu ratio (SRV < LF < BSC).

This additional result partially agrees with some differences in semen quality parameters found among the three subgroups of donors in Montano et al. (2021) [26].

This is in line with our recent studies, which have shown that copper, in the oxidation state 2+, interacts with arginine residues of sperm H1 histones (particularly rich in arginine), inducing oxidative DNA damage in the presence of hydrogen peroxide [20]. Moreover, Human Protamine 2 has a strong Cu(II)-binding amino acid motif at its N-terminus (Arg-Thr-His), which is able to mediate oxidative DNA double-strand scission and the generation of 8-oxo-20-deoxyguanosine (8-oxo-dG) from free 20-deoxyguanosine (dG) and from DNA by H_2_O_2_ [88,89]. Considering this evidence, and also that sperm DNA fragmentation is one of the primary causes of male infertility, it would be possible to hypothesize that these proteins could trap this metal, increasing the availability of Cu(II) ions near the binding surface of DNA. Consequently, this condition could lead to the promotion of the Fenton reaction in DNA proximity after H_2_O_2_ addition, determining DNA breakage and explaining the DNA oxidative damage found in spermatozoa of men living in polluted areas and presenting a high level of copper in their semen [18,19].

Furthermore, this study provides a starting point for ongoing work to evaluate the effect of a dietary intervention on the profile of trace elements in serum and semen. If differences between the areas are not eliminated after dietary intervention, they may be more closely related to environmental exposure.

## 5. Conclusions

In conclusion, the current results show the levels of 26 elements in both human semen and blood serum of a population of Italian individuals who are all healthy, young men aged 18 to 22 years, homogeneous for age, anthropometric characteristics, and lifestyle, living in three environmentally polluted geographic areas. These differences could be due to different dietary intake and environmental exposure in the areas of recruitment. Nevertheless, more research is needed to evaluate threshold values for sperm dysfunction and male infertility. According to our findings, the determination of the possibly hazardous elements, simultaneously, in human semen and blood serum, could be useful to speculate some correlation with environmental pollution and dietary intakes. The bioaccumulation phenomenon of some trace elements in semen where inter-area differences are more evident in semen than in blood serum should be further investigated in order to eventually identify some of them as novel biomarkers of environmental exposure. This work suggests human semen as an early environmental marker. It also underlines the need for further detailed analysis of the pollution sources by the competent authorities.

## Figures and Tables

**Figure 1 ijerph-19-11635-f001:**
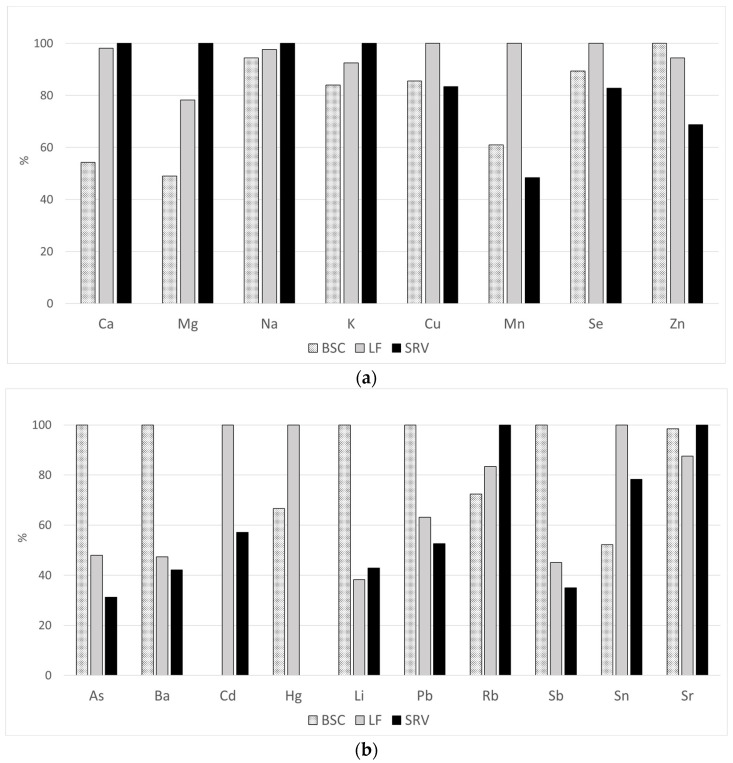
Percentage Elemental Relative Concentrations in blood serum. For each element, the percentage calculated with formula (1) of each group (BSC, LF, SRV) is reported. (**a**) shows the elements defined as macro and essential ones; (**b**) shows the elements defined as additional and non-essential ones.

**Figure 2 ijerph-19-11635-f002:**
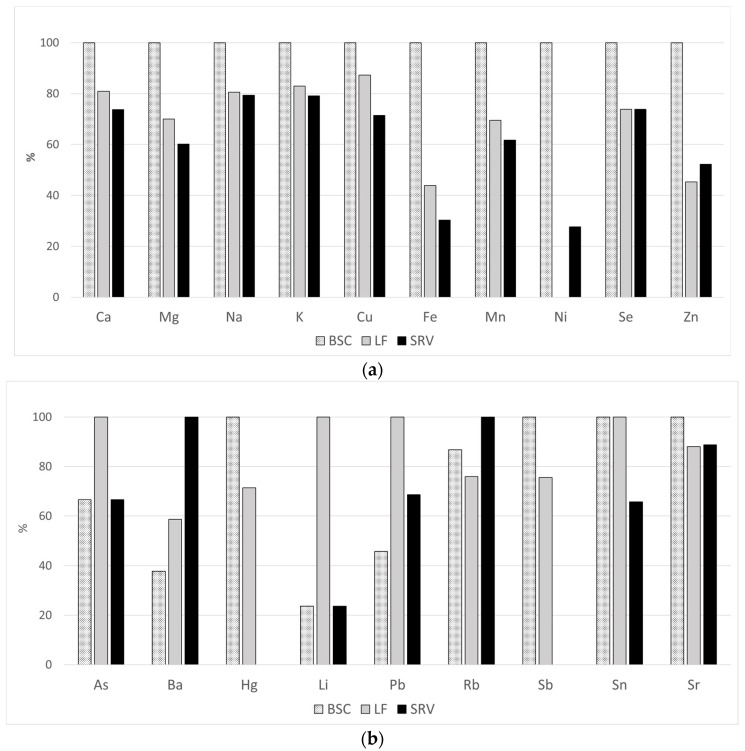
Percentage Elemental Relative Concentrations in semen. For each element, the percentage calculated with formula (1) of each subgroup (BSC, LF, SRV) is reported. (**a**) shows the elements defined as macro and essential ones; (**b**) shows the elements defined as additional and non-essential ones.

**Table 1 ijerph-19-11635-t001:** Concentration of Group II—macro elements (mg/L) in blood serum and semen samples of the whole cohort (WRC) and three areas (BSC, LF, SRV), reported as median values and ranges.

		WRC			BSC			LF			SRV		
	*n*	Median	Range	*n*	Median	Range	*n*	Median	Range	*n*	Median	Range	*p* Value ^1^
Blood serum													
Ca	330	157	52.4–4627	144	102	52.4–251	135	184	116–4627	51	188	145–3082	0.0001
Mg	330	29.8	10.7–135	144	20.1	10.7–44.1	135	32.1	22.7–119	51	41	33.4–135	0.0001
Na	256	4455	3238–10,449	70	4322	3238–6399	135	4471	3485–7383	51	4576	4115–10,449	0.0001
K	256	229	123–878	70	210	123–278	135	231	162–878	51	250	200–570	0.0001
Semen													
Ca	262	488	117–1193	113	560	237–1193	94	453	133–1171	55	412	117–773	0.0001
Mg	262	128	11.6–449	113	172	25.6–441	94	120	11.6–449	55	103	19.7–251	0.0001
Na	262	4302	91.9–9745	113	4915	2397–9745	94	3960	91.9–7777	55	3900	354–6759	0.0001
K	262	1866	12.7–3879	113	2063	1074–3879	94	1712	12.7–3464	55	1632	145–3508	0.0001

*n*—number of donors; ^1^*p*-values were calculated with the Kruskal–Wallis Test

**Table 2 ijerph-19-11635-t002:** Concentration of Group III—essential trace elements (µg /L) in blood serum and semen samples of the whole cohort (WRC) and three areas (BSC, LF, SRV), reported as median values and ranges.

		WRC			BSC			LF			SRV		
	*n*	Median	Range	*n*	Median	Range	*n*	Median	Range	*n*	Median	Range	*p* Value ^1^
Blood serum													
Cu	332	842	445–2049	144	786	461–2049	137	918	554–1807	51	766	445–1103	0.0001
Mn	286	4.8	0.4–35.9	124	3.9	0.4–19.5	137	6.4	1.4–35.9	25	3.1	0.8–13.1	0.0001
Se	332	103	50.3–248	144	98.4	66.5–248	137	110	85.7–162	51	91.1	50.3–122	0.0001
Zn	331	1204	618–4820	144	1285	736–4820	136	1214	710–2026	51	884	618–1358	0.0001
Semen													
Cu	268	142	36.9–1085	113	165	76.6–635	100	144	44.1–1085	55	118	36.9–482	0.0001
Fe	268	1375	329–119,401	113	2662	428–47,180	100	1166	329–119,401	55	807	375–88,138	0.0001
Mn	268	10.1	2.5–133	113	12.8	3.3–108	100	8.9	2.5–133	55	7.9	2.7–50.6	0.0033
Ni	152	14.4	4.4–240	109	27.1	5.5–240	100	<4.2	-	43	7.5	4.5–66.8	-
Se	268	38.2	3.9–119	113	49.4	18.4–110	100	36.5	6.6–119	55	36.5	6.6–119	0.0001
Zn	268	91,316	48–526,312	113	130,430	400–526,312	100	59,137	48–219,654	55	68,206	200–223,536	0.0001

*n*—number of donors; ^1^*p*-values were calculated with the Kruskal–Wallis Test

**Table 3 ijerph-19-11635-t003:** Concentration of Groups IV and V—additional and non-essential trace elements (µg /L) in blood serum and semen samples of the whole cohort (WRC) and three areas (BSC, LF, SRV), reported as median values and ranges.

		WRC			BSC			LF			SRV		
	*n*	Median	Range	*n*	Median	Range	*n*	Median	Range	*n*	Median	Range	*p* Value ^1^
Blood serum													
As	332	2.9	0.2–97.9	144	4.8	0.3–36.9	137	2.3	0.7–97.9	51	1.5	0.2–13.7	0.0001
Ba	261	18.0	6.7–1268	129	25.6	6.7–1268	103	12.1	6.8–145	29	10.8	6.8–27.4	0.0001
Cd	105	0.7	0.4–2.7	144	<0.2	-	86	0.7	0.4–2.7	19	0.4	0.2–1.3	-
Hg	144	0.7	0.2–4.4	95	0.6	0.2–2.9	88	0.9	0.2–4.4	51	<0.2	-	-
Li	317	12.5	0.7–371	136	30.3	0.7 – 106	132	11.6	1.8–22.3	49	13	8.3–371	0.0295
Pb	318	1.3	0.1–231	136	1.9	0.2–231	131	1.2	0.1–40.5	51	1	0.2–3.9	0.0001
Rb	332	159	84.2–1093	144	141	89.4–334	137	162	91.3–1093	51	194	102–289	0.0001
Sb	292	1.1	0.2–7.6	137	2	0.3–7.6	113	0.9	0.3–6.7	42	0.7	0.2–7.2	0.0001
Sn	281	1.9	0.2–36.9	96	1.2	0.2–36.9	136	2.3	0.2 – 5.1	49	1.8	0.2–3.6	0.0001
Sr	332	30.8	14.5–124	144	32.3	15.1–124	137	28.7	14.5–60.3	51	32.8	14.7–56.0	0.0019
U	193	0.3	0.2–2.5	72	0.3	0.2–2.5	96	0.3	0.2–1.5	25	0.3	0.2–0.7	0.0764
Semen													
As	267	4.6	0.2–33.7	112	4.0	0.2–33.7	100	6.0	1.6–17.8	55	4.0	1.0–16.5	0.0001
Ba	204	74.5	26.0–19,847	49	48.4	26.0–103	100	75.2	27.6–3177	55	128.2	48.3–19,847	0.0001
Hg	120	0.5	0.2–2.7	39	0.7	0.2–1.8	81	0.5	0.2–2.7	54	<0.2	-	-
Li	267	27.5	0.4–210	112	24.2	0.4–43.5	100	102	1.4–210	55	24.2	5.2–35.5	0.0001
Pb	257	2.4	0.1–48.7	102	1.6	0.1–11.5	100	3.5	0.4–22.4	55	2.4	0.5–48.7	0.0001
Rb	268	1489	214–4847	113	1504	688–2717	100	1315	328–3053	55	1733	214–4847	0.0025
Sb	100	3.9	0.2–79.4	66	4.1	0.2–14.1	34	3.1	0.3–79.4	55	<0.2	-	-
Sn	260	3.5	0.2–28.4	112	3.5	3.1–27.4	94	3.5	2.4–28.4	54	2.3	3.6–8.8	0.0001
Sr	268	70.8	206–272	113	76.1	32.3–272	100	67.0	22.3–186	55	67.6	20.6–152	0.008

*n*—number of donors; ^1^*p*-values were calculated with the Kruskal–Wallis Test

## Data Availability

Not applicable.

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
