# Peer review of "Comparison between Macro and Trace Element Concentrations in Human Semen and Blood Serum in Highly Polluted Areas in Italy"

_ijerph, 2022, doi:10.3390/ijerph191811635_

Round 1

Reviewer 1 Report

This is a good piece of work. Congratulations!

I have recommended some extremely minor optional language  edits ( see comment bubbles on the attached document).  

Author Response

We thank the reviewer very much for appreciating our work.

We have made all the optional language changes suggested.

We also attached pdf with the responses to the comments in the bubbles.

Reviewer 2 Report

Overall, it is a very important article, however, references should be updated.

For example, line 75, reference 10, mentions in recent years referring to 2012.

The description of results and methodology is correct. I suggest to improve the quality of figures 1 and 2. The conclusions could be more specific to what was found in the study.

Reviewer 3 Report

The study is very interesting but the result should be rephrased again by a summarized form the result writing is so bad.....many sections should be reorganized again such as discussion and conclusion as those sections are so long need to be full rewritten....what about the study approval number? What about study limitations?? In materials and methods what about inclusion and exclusion criteria???all paper items should be reorganized 

Reviewer 4 Report

Title: Comparison between macro and trace element concentrations in human semen and blood serum in highly polluted areas in Italy

COMMENTS:

Line 113:

Comment: is there an ethics number(s) for this study?

Line 153:

Comment: please add more detail on how/where blood samples were collected

Line 147: 

Comment: why was there no control group?  i.e. a population of males from a 'non-polluted' area?

Line 180: "The elements of Group I were not object of this study."

Comment: then why include them in the methods?  Would be simpler to just outline 4 groups?

Line 183: "Iron was evaluated only in semen samples"

Comment: not sure I understand why serum iron wasn't also evaluated

Line 195: "The difference in the number of analyzed samples was because it was not possible to perform all determinations on all samples."

Comment: how was it decided to use which sample for which analysis?  Randomly allocated?  if so please state.

Line 197: "A blank from collecting tubes used in the different areas"

Comment: not sure what this means?

Line 205: "The results of the test samples were reported considering the values obtained for each element in the blank samples."

Comment: do you mean the data was indexed to the blank sample for each area? how was this done?

Line 231: "in a database."

Comment: what kind of database?  excel file?

Line 263: Tables1-6

Comment: You divide the elements into separate groups (I-V) in the methods, but don't refer to these groups in the results? Should at least include group numbers in table legends.

Comment: extremely hard to interrupt any of these tables without a control sample group.  i.e. concentrations of these elements in semen and serum of subjects living in a non-polluted area.  Significant differences feel moot without them.  Maybe include reference values in these tables (such as presented in Tables S9, S10).  Would much prefer seeing reference values over WRC values.

Comment: suggest combining semen and serum tables.  i.e. Tables 1+2, Tables 3+4, Tables 5+6.

Line 305: Table 4

Comment: Missing Ni and Fe.  Still not clear why Fe was excluded.

Line 321: Table 5. 

Comment: missing values for Cd and U? These were mentioned in text?

Line 350: Equation 1.

Comment: calculating relative abundances of these elements seems pointless without a true control population value?  Would be much clearer if results were compared to normal reference values (such as presented in Tables S9, S10)

Line 360; 371: Figures 1-2

Comment: This is basically the same data as in tables 1-6 being presented again.

Comment: was correlation analysis possible between trace element abundance and sperm quality? (As presented in table S2) Was sperm quality assessed for the same subjects or just subjects from the same area?  Methods weren't clear.

SUGGESTED EDITS

Line 27: suggest change "main regulators" to "important regulators"

Line 33: suggest change " a comparison among the three areas was performed" to " a comparison among three geographical areas was performed"

Line 49: suggest change "The elements are present in living tissues in several amounts and in different forms. Some important biological processes are regulated by trace elements. To better evaluate elements’ functions, several classifications have been proposed, according to specific features and considering them essential for the normal development and growth. For instance, they can be easily categorized in two groups, macro and trace elements [1–3]" to

"Varying concentrations of different minerals, known as trace elements, are found in the body and are essential to maintain normal growth and development of living tissues [1–3]."

Line 54: After above sentence suggest including some text outlining some common trace elements and what role they play in normal physiology.

e.g. "Examples of essential trace elements include minerals such as Iron (Fe), chromium (Cr), copper (Cu), zinc (Zn), selenium (Se), molybdenum (Mb), cobalt (Co), and iodine (I).  While these elements only account for 0.01% - 0.02% of the total body weight [Citation], they play significant roles such as enzyme catalysts or in oxidation-reduction reactions for energy metabolism [Citation].  Iron in particular is a vital component in haemoglobin and myoglobin for the transport of oxygen [Citation]."

Line 54: suggest change "Then, the necessity/requirement to set a level for daily intake arises to meet physiological needs and, thereby, to reduce the risk of nutrient deficiency or excess. Furthermore, for the known essential elements, essentiality and toxicity are two sides of the same medal. Hence, toxicity of a trace element is a matter of dose. All trace elements can be potentially toxic if consumed at high levels for prolonged periods and the toxicity is expressed when the element’s concentration brings about an impairment in physiological functions and/or the onset of pathological manifestations. These trace elements could have clinical significance and might be analytically evaluated [4,5]." to

"All trace elements can be potentially toxic if consumed at high levels for prolonged periods, which can result in impairment of normal physiological functions and/or the development of pathological conditions [4,5]."

Line 62: suggest change "The concern about the role of trace elements as potential toxicants on human health has arisen due to their persistence combined with the possible bioaccumulation, and their extensive use in various industrial processes. It is well known the harmful impact of some metals on living beings. Indeed, it is not by chance that the first three positions of the Agency for Toxic Substances and Drug Registry (ATSDR) 2019 Substance Priority List [6] are occupied by three elements, arsenic (As), lead (Pb) and mercury (Hg), respectively." to

"The potential for trace elements to act as toxicants on human health is of particular interest due to their extensive use in various industrial processes which can result in their release into the environment resulting in bioaccumulation and poisoning within our bodies.  Trace elements found with increasing frequency in nature with recognised harmful impacts on living beings include elements such as arsenic (As), lead (Pb) and mercury (Hg) [6]."

Line 68: suggest change "Toxicity of elements is correlated to" to "The toxicity of trace elements is correlated with"

Line 69: suggest change "biotransformation. They are all host-based factors that can affect the toxicity mechanism [7]." to "biotransformation, which are all host-based factors that can affect the toxicity mechanism [7]."

Line 71: suggest change "intoxication/exposure. For instance, cigarette smoke by itself" to "intoxication/exposure.  Cigarette smoke itself"

Line 72: suggest change "and it is referred as the main " to "and is considered the main "

Line 74: suggest change "In the last years, the human male reproductive system has faced an increasing incidence of disorders worldwide [10]." to

"The incidence of human male reproductive disorders is increasing worldwide [10]."

Line 77: suggest change "In particular, the increased number of fertility disorders and the widespread exposure of the general population to metalloid and metal species underlined a potential correlation [13,14]." to

"The increased number of fertility disorders is potentially correlated with the widespread exposure of the general population to metalloid and metal species [13,14]."

Line 81: suggest change "cop-per" to "copper"

Line 85: suggest change "though ROS are" to "though reactive oxygen species (ROS) are"

Line 85: suggest change "Hence, the use of human semen as an early and sensitive biomarker of environmental exposures to pollutants may lead to a better comprehension of how the environmental toxicants can affect the healthy humans [13,20,21]." to

"Hence, human semen is seen as an early and sensitive biomarker of environmental exposure to pollutants which may lead to a better understanding of how environmental toxicants can affect healthy humans [13,20,21]."

Line 89: suggest change "The aim of this study was to evaluate blood serum and semen levels of trace elements in healthy young men homogeneous for age, anthropometric characteristics and lifestyle, living in three highly polluted areas in Italy in order to understand whether the investigated parameters could allow discriminating among these three groups, and possibly verify how much the environmental conditions of the recruitment areas can play a role in the bioaccumulation rate of the trace elements." to

"The aim of this study was to evaluate blood serum and semen levels for trace elements in healthy young men homogeneous for age, anthropometric characteristics and lifestyle, living in three highly polluted areas in Italy; and determine how the environmental conditions of the recruitment areas influence the bioaccumulation of these trace elements."

Line 96: suggest change "ICP-MS" to "Inductively coupled plasma mass spectrometry (ICP-MS)"

Line 99: suggest change "For this purpose, the subjects recruited " to "Subjects recruited "

Line 99: suggest change "participants of a " to "participants of the "

Line 119: suggest change "Montano et al (2020)" to "Montano et al (2021)"

Line 119: suggest change "A short overview is provided here. The " to "In brief, the "

Line 123: suggest change "All recruited subjects got informed of" to "All recruited subjects were informed of"

Line 124: suggest change "at the enrollment" to "at enrollment"

Line 126: suggest change "from 18 to 22 years old for subsequent analysis" to "aged between 18 to 22 years old."

Line 127: suggest change "for the recruitment fall between the " to "for subject recruitment were located between the "

Line 128: suggest change "contaminated" to "polluted"

Line 133: suggest change "Samples from the first site (BSC, n = 142) came from the SIN called Brescia-Caffaro" to 

"Samples from the first site came from the SIN called Brescia-Caffaro (BSC, n = 142), "

Line 136: suggest change "which produced PCBs" to "which produced Polychlorinated Biphenyls (PCBs)"

Line 138: suggest change "Samples of the second area (SRV, n = 54) came from the Sacco River Valley, " to

"Samples of the second area came from the Sacco River Valley (SRV, n = 54), "

Line 139: suggest change "located nearby Rome, heavily polluted" to "located near Rome, which is heavily polluted"

Line 140: suggest change "from chemical industrial" to "from a chemical industrial"

Line 143: suggest change "Samples of third group (LF, n = 127) came from the municipalities belonging to the 143 “Land of Fires”," to

"Samples from a third area came from the municipalities belonging to the “Land of Fires” (LF, n = 127),"

Line 145: suggest change "involved in toxic waste scandal" to "know for illegal waste dumping and toxic fires"

Line 154: suggest change "An aliquot of 1 mL " to "A 1 mL aliquot of serum"

Line 158: suggest change "To better evaluate " to "To evaluate "

Line 158: suggest change "To better evaluate differences in blood serum and semen, we divided the comparisons among the detected chemical elements according to the classification used by WHO, which considers the nutritional importance and the daily requirement, divided the elements in five groups [1,3]." to

"To better evaluate differences in blood serum and semen we grouped detected chemical elements into five groups according to the classification used by WHO, which considers element nutritional importance and its daily requirement [1,3]."

Line 162: suggest change "The Group I" to "Group I"

Line 164: suggest change "The Group II is" to "Group II was"

Line 167: suggest change "The Group III is" to "Group III was"

Line 167: suggest change "elements, as" to "elements, such as "

Line 170: suggest change "The Group IV is" to "Group IV was"

Line 170: suggest change "elements, that is elements with a not well-defined role, probably essential, as " to

"elements, that includes elements with a not well-defined role but which are most likely essential, such as "

Line 177: suggest change "The Group V is" to "Group V was"

Line 178: suggest change "toxicity, as aluminum" to "toxicity, such as aluminum"

Line 204: suggest change "were run" to "was run"

Line 206: suggest change "quanti-fica-tion" to "quantification"

Line 210: suggest change "The precision of described method was estimated by performing at least seven replicates on unfortified blood serum sample. The sample was fortified by known additions (spikes) of elements not contained in unfortified sample." to

"Trace element measurement precision was estimated by performing at least seven replicates on an unfortified blood serum sample. The sample was then fortified by known additions of elements not contained in the unfortified sample."

Lines 246: suggest change "We show below the data obtained on the whole recruited cohort (WRC) and the values obtained in the three subgroups of donors from Brescia-Caffaro (BSC) in Lombardy Region, Sacco River Valley (SRV) in Lazio Region and “Land of Fires” (LF) in Campania Region. The general features of the recruited young men living in the three areas are reported in Table S1. General characteristics of the study population reported for each area as mean value ± standard deviation. The three groups were similar for body weight, height, BMI and abdominal circumference; the BMI in some boys was higher than the limit of 25 for normal weight but in this case, they were subjects with particularly developed muscles, and it was not due to abdominal obesity as assessed with the measurement of abdominal circumference. Only the average age is higher in BSC group." to

The general characteristics of the young men recruited from the three polluted areas were similar for body weight, height, BMI and abdominal circumference (Table S1); the BMI in some boys was higher than the limit of 25 for normal weight but in this case, they were subjects with particularly developed muscles, and it was not due to abdominal obesity as assessed with the measurement of abdominal circumference. Only the average age was higher in the BSC group (Table S1).

Line 259: suggest change " of the elected population were widely" to " of the subject groups was previously"

Round 2

Reviewer 3 Report

The version now is modified so I accept it

Reviewer 4 Report

I thank the authors for their thorough edits and answers to my original comments. 

While I'm still not convinced that their lack of control group (i.e. a population of males from a 'non-polluted' area) is warranted.  I find the data of sufficient interest to be published "as is".